# CO$_2$ Emissions from Plastic Consumption Behaviors in Thailand

**Virin Kittithammavong * , Wilawan Khanitchaidecha and Pajaree Thongsanit**

Department of Civil Engineering, Naresuan University, Phitsanulok 65000, Thailand; wilawank@nu.ac.th (W.K.); pajareet@nu.ac.th (P.T.)
* Correspondence: virink@nu.ac.th

**Abstract:** Plastic waste is an environmental crisis that is becoming increasingly well-documented. The rapid expansion of plastic manufacturing and consumption has led to a harmful cycle of pollution and greenhouse gas emissions due to petroleum-based production and plastic waste disposal. Plastic production and disposal depend on the consumption behavior of people. This study aimed to examine the plastic consumption behavior in Thailand and its impact on climate change at the end-of-life stage. The general information, plastic consumption, and plastic waste management were collected via questionnaires for each product lifetime, including single-use, medium-use, and long-use plastics. Based on 567 questionnaires, the results showed that people consumed single-use plastic, e.g., plastic bag, food container, cutlery, straws, and bottles, at a rate of about nine pieces/household/day or three pieces/cap/day. The medium-use and long-use plastic were 10 pieces/household/month and 50 pieces/household/year, respectively. It should be remarked that population density, education, and number of household members affected plastic consumption behavior, especially for single-use plastic. Regarding the disposal of end-of-life plastics, Thai people, on average, contribute 0.15 kg CO$_2$eq/household/day to climate change. Many households have mismanaged waste by open dumping and open burning. Therefore, practicing proper waste management will help Thailand on the path to carbon neutrality in the future.

**Keywords:** municipal solid waste (MSW); waste management; greenhouse gas emissions; climate change; circular economy

## 1. Introduction

Plastics are essential raw materials for daily commodities, such as food containers, cutlery, clothes, and appliance components. Before the COVID-19 pandemic, global plastic production was around 370 million tons per year. Then, it increased to 390.7 million tons in 2021 [1]. The product manufacturers and consumer behaviors drive the demand for plastics. Lightweight flexibility is a unique property of plastics. Many industries have used plastics to reduce product weights and enhance the possibility of a product design [2]. In Thailand, more than 34.3 million tons of plastics were produced for daily-life products in 2022 [3]. In total, 96.7% of them were used for shoe manufacturing. Other applications included bags, pipes and joints, packaging, color, film, sheet, big bags, and tableware and toilet ware, respectively. Most plastic products are fossil-based products [1]. The general types of plastics are polyethylene terephthalate (PET), low-density polyethylene (LDPE), high-density polyethylene (HDPE), polypropylene (PP), polystyrene (PS), and polyvinyl chloride (PVC), which can be recycled [1,4].

Nowadays, global plastic use has grown exponentially, as well as plastic production. In 2016, the world plastic consumption rate was 45 kg per capita per year (kg/cap/year), or 0.12 kg/cap/day [5,6]. North America had 139 kg/cap/year, which was the highest plastic consumption in the world. Europe had 48–136 kg/cap/year because Western and Eastern Europe have a difference in production and waste policies. Asia, with the exclusion

of Japan, showed the lowest consumption, at 36 kg/cap/year (0.10 kg/cap/day), which is lower than the world average. Plastic consumption in developed countries is higher than in developing countries. People in developed countries have more chances for product and service access. The per capita consumption of plastics in Thailand was 64 kg/year or 0.18 kg/cap/day in 2017 [7]. It was higher than the world's average consumption but lower than that in North America and some parts of Europe. Thailand is responsible for about 9% of the world's plastic production and is one of the top three plastic-packaging-producing countries in Southeast Asia [7].

Plastic products can be divided into various categories based on lifetimes. Wang et al. (2019) divided plastics into four lifespans: 1–2 years, 3–5 years, 6–9 years, and more than 10 years [5]. Polyethylene, polypropylene, and polystyrene primarily belong to the short-lifespan group (less than 2 years), while polyvinyl chloride usually has a longer lifespan. Plastics are likewise classed as durable and non-durable plastics. Durable plastics are applied to vehicles, electronic equipment, building materials, and appliances. Their lifetime is generally designed to be 5–50 years or more [8]. The non-durable plastics are used for daily-life products such as cloths, containers, and tableware. Typically, plastics fall into the categories of short (trash bags, single-use containers, and diapers), medium (fabrics, furniture, and small tools), and long (automobiles, building materials, and large equipment) lifespans [6,8].

People dispose of plastic waste as municipal solid waste (MSW). In 2019 only 9% of plastic waste was recycled globally. In total, 69% ended up in landfills and incinerated, while 22% were mismanaged [9]. One of the highest plastic waste-generating countries is the US, which produces 42 million metric tons of plastic waste and 130 kg/cap/year [6]. About 86% of that plastic waste is disposed of in landfills, 9% is transferred to incineration sites, and only 5% is recycled. In Thailand, plastic waste makes up around 12–18% of the total MSW each year [7,10]. In addition, 21% of the total MSW is mismanaged through open dumping, open burning, and being disposed of in an unwell small incinerator [7]. Mismanagement leads to negative environmental impacts, e.g., air pollution, water pollution, ecosystem impacts, hazardous chemical emissions, microplastic contamination, and greenhouse gas (GHG) emissions [11]. GHG emissions are especially an issue around the world in the face of climate change. In the European Union (EU), the life cycle of plastics resulted in 208 million tons of carbon dioxide equivalent ($MtCO_{2eq}$) in 2018 [12]. The end-of-life stage caused 15% of total emissions or around 31 $MtCO_{2eq}$. Thailand showed that the GHG emissions of the waste sector accounted for 11.43–16.77 $MtCO_{2eq}$ (3.74–4.73% of the total GHG emissions) [13,14]. This sector has been one of the targets for reducing emissions in the country. However, $CO_2$ emissions from plastic consumption have not been proposed.

Nowadays, international agreement advocates for the limitation of the global average temperature caused by GHG emissions to be limited to 1.5–2 °C [15]. The cumulative $CO_2$ emission should not exceed 1000 $GtCO_2eq$ during the period from 2000 to 2050. However, a quantity of 234 $GtCO_2eq$ was released and accrued in the atmosphere between 2000 and 2006 [16]. The Organization for Economic Cooperation and Development (OECD) reported the significant role of plastic in contributing to GHG emissions, with the complete life cycle of plastic products responsible for emitting 1.8 gigatons of carbon dioxide, accounting for 3.7% of global emissions in 2019 [17]. It is projected to increase to 4.3 gigatons of carbon dioxide (4.5% of global emissions) in 2060 and could account for more than 19% of the total carbon budget in the world [18]. Thus, plastic is one of the big sources of GHG emissions, contributing to the escalation of global temperatures. The consequences of surpassing the 1.5–2 °C warming threshold are manifold, encompassing a range of risks to both natural and human systems, including rising sea levels, diminished ecosystems, altered land use, food security loss, and potential heatwaves [19]. Many cases showed the adaptive response of the body size of creatures to survive climate change [20]. For example, the alpine (*Viola biflora*) reacts to climate change by changing the leaf size and number of leaves [21]. Notably, if global warming were to exceed 4 °C, the alpine species

could face a mortality rate ranging from 90% to 100%, as found in the study by Cui et al. (2018) [21].

Thailand is an industrialized and rapidly developing country, which has led to an increase in greenhouse gas emissions. The country's primary contributors to GHG emissions include the energy sector, agriculture, industry, and waste management [14]. Over the past two decades, Thailand's total GHG emissions have increased substantially, from 246 MtCO$_2$eq in 2000 to exceeding 350 MtCO$_2$eq presently [14]. These emissions have also increased Thailand's annual mean maximum temperature. Thailand has faced severe weather events, such as storms, floods, and droughts, resulting in economic damages. For example, after the 2011 flood, 65 out of 77 provinces were declared disaster areas. This catastrophic event resulted in extensive damage to the agricultural, industrial, and residential sectors, with an estimated cost of approximately 45.7 billion US dollars [22].

Many countries have launched policies, campaigns, and roadmaps to reduce plastic waste generation and its impacts. Thailand instituted the "Roadmap on Plastic Waste Management 2018–2030", which aims to reduce the use of single-use plastic by replacing it with environmentally friendly products and to return 100% of the target plastic waste to the circular economy [10]. Oxo-degradable plastic, cap seals, and plastic microbeads were banned in 2019. By 2070, all plastic waste in Thailand is expected to be 100% recycled. However, the roadmap cannot be completed without people's awareness. Behavior changes are vital to reducing plastic waste sustainably [11]. Many studies have focused on plastic consumption behaviors [23–29]. One found that students in Indonesia consumed plastic bottles at 1.39 g/cap/day, plastic bags at 0.14 g/cap/day, and plastic cups at 0.20 g/cap/day [28]. Another study showed that most Indonesians (43.18% of total respondents) used 2–4 pieces/cap/day of plastic bottles, and 36.36% used plastic bottles at more than four pieces/cap/day [24]. Similarly, a study of English people found that most respondents (67% of the total sample) consumed plastic bottles at about 1–5 pieces/cap/day [27]. The factors affecting plastic consumption behaviors included gender, age, education, income, and community engagement [23,25,26,29]. For example, women tended to "think green" by carrying their own bags or reusing plastic bags. People with higher incomes preferred bringing their own bags, while those with lower incomes preferred to use convenient plastic bags. In addition, external factors such as green policies and campaigns can drive public awareness [23,25]. Environmental concern encourages customers to reduce plastic consumption and promote sound waste management [29].

Therefore, this research aimed to investigate the plastic consumption behavior of people in Thailand through questionnaires. The study collected and analyzed demographic data such as municipality type, residential activity, education, household size, and household income to explore their relationship with the quantity of plastic consumption. Quantitative measures of plastic consumption and waste management were obtained and subsequently categorized into three groups: single-use, medium-use, and long-use plastics. Moreover, the study evaluated the greenhouse gas emissions of end-of-life plastics. The quantitative measurements of consumption and emissions hold potential value for devising more effective strategies for plastic management.

## 2. Methodology

### 2.1. Questionnaire Design

This study utilized a survey method to gather data. Questionnaires were distributed throughout all regions in Thailand. Google Forms was employed as a tool for participants to respond to the survey. The questionnaires were structured into two main sections: demographic information and plastic consumption behavior. Demographic data included gender, age, address, municipality type, residential activity, education level, household size, and household income. Municipality type refers to population density, including special administrative areas (Bangkok and Pattaya), city municipality (over 50,000 inhabitants), town municipality (15,000–50,000 inhabitants), subdistrict municipality (7000–15,000 inhabitants), and subdistrict administrative organization (fewer than 7000 inhabitants). Residential

activities were categorized into four types: agriculture, commerce, industry, and tourism. The study explored the relationship between plastic consumption and variables such as municipality type, residential activity, education level, household size, and household income. The section on plastic consumption behavior collected data on short-, medium-, and long-lifespan plastics. Short-lifespan plastics refer to single-use items discarded after a brief period (within a week). Single-use plastics, such as plastic bags, food bags, and straws, were also studied. Medium-lifespan plastics, such as detergent bottles, shampoo bottles, and toothbrushes, showed longer lifespans than single-use plastics. Additionally, long-lifespan plastics, such as reusable bottles, bags, and durable household items, such as shelves, tables, and kitchenware, were examined. Quantitative data (e.g., pieces per day, pieces per year) and qualitative information (i.e., waste management) were collected for these three types of plastics. Large objects, such as vehicles, electrical equipment, and building materials, were excluded from the study. Figure 1 shows the plastic products that were considered in this study.

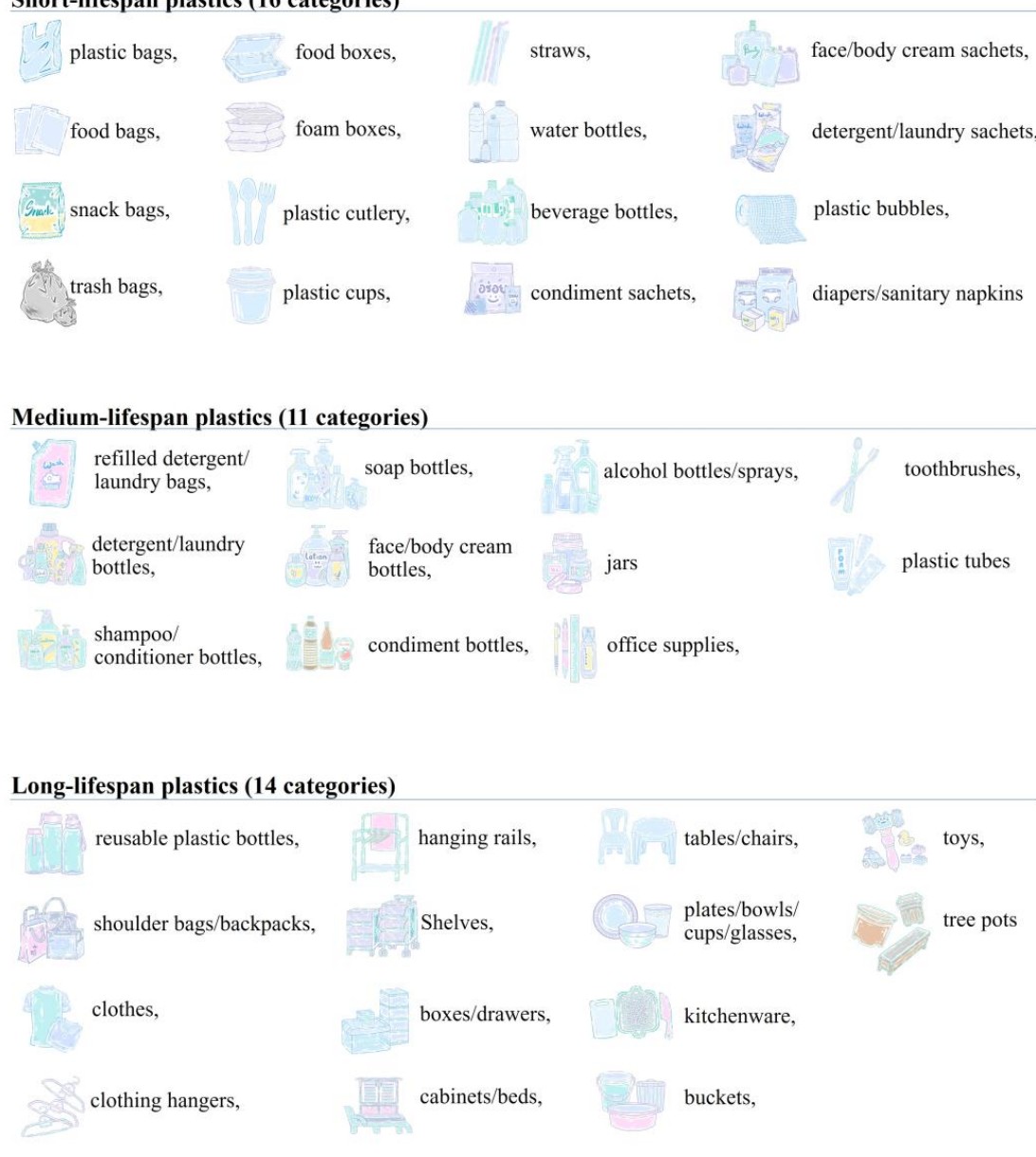

**Figure 1.** Plastic products considered in this study.

### 2.2. Sample Size

The target sample size was determined using Taro Yamane's (1967) equation, which reflects a general random sampling technique. The formula is shown in Equation (1):

$$n = \frac{N}{1 + N(e)^2} \tag{1}$$

where n = sample size, N = Thailand's population (66,090,475 people based on 2022 census data [30]), and e = precision or sampling error (5%). Therefore, the target sample size was 400, and at least 400 questionnaires would be collected.

### 2.3. Data Analysis

For the data analysis, a multiple linear regression was employed as a statistical model to examine the relationship between plastic consumption and the demographic variables. A regression analysis tool in Excel (Microsoft 365 MSO Version 2307) was used to perform ANOVA. The significance F (Sig-F) and *p*-values were considered acceptable at 0.05 or less. The results showed Sig-F and *p*-values $\leq$ 0.05, meaning the variables were related linearly.

### 2.4. Greenhouse Gas Emissions Analysis

The end-of-life plastic products' greenhouse gas (GHG) emissions were determined using ISO14040 [31]. The GHG emissions were calculated using Equation (2):

$$\text{GHG emissions} = \text{Activity data} \times \text{Emission factor} \tag{2}$$

where GHG emissions = kilogram carbon dioxide equivalent ($kgCO_2eq$), activity data = amount of plastics (kg), and emission factor (EF) = GHG emissions per unit ($kgCO_2eq/kg$).

Typically, Thailand has four approaches to municipal waste management, as listed in Table 1. First, trash is disposed to public waste containers. The local municipalities are responsible for collecting, treating, and disposing of waste in their districts. However, the waste management system does not reach some areas. Here, residents rid of their trash by dumping and covering or openly burning it in their residential areas. Some valuable plastics can be sold at scrap shops located throughout the country. The emissions per unit of end-of-life plastic products are presented in Table 1. The recycling process was assumed to be negligible because the waste could be repurposed as raw material in various industries [32]. Preceding their sale at junk or scrap shops, plastic wastes are initially segregated at the household level. Subsequently, the scrap shops collect and prepare these materials for transfer to the recycling manufacturers. In this study, emissions related to recycling were identified from the household separation to the scrap shop gate, with the emission considered zero due to the exclusion of transportation-related impacts from the analysis.

**Table 1.** Emissions per unit of end-of-life plastic products.

| Municipal Waste Management | Emission Per Unit (kgCO$_2$eq/kg Plastic Product) | Remark |
|---|---|---|
| Disposal to public container | 0.79 | Thailand Greenhouse Gas Management Organization (TGO), 2023 [33] |
| Dumping and covering | 1.04 | Thailand Greenhouse Gas Management Organization (TGO), 2023 [33] |
| Open-burning | 3.54 | Calculation based on TGO guidelines (2016) and GWP AR5 2014 [34] |
| Recycle (sell to junk shop/scrap shop) | 0 | Technical guidance for calculating Scope 3 emissions [32] |

Furthermore, the analysis of GHG emissions was delimited to encompass only the end-of-life phase of plastic products, excluding emissions resulting from fuel utilization during waste collection and transportation to final disposal sites.

## 3. Results and Discussion

### 3.1. Plastic Consumption Behaviors

The questionnaires were obtained between January and March 2023, yielding 567 responses. Each respondent provided information on behalf of a single household. Figure 2 presents the demographic details of the respondents. Most participants were female, comprising 62% of the sample, while males accounted for 37%; the remaining respondents did not specify their gender. Respondents' ages ranged from 13 to 69 years, and 73% of respondents were between the ages of 21 and 40 years. According to respondents' addresses, data collection encompassed 61 out of 77 provinces in Thailand, with the highest concentration of respondents residing in Bangkok (35%), followed by Nonthaburi (11%) and Ranong (7%). The participants represented a diverse range of municipal areas. The questionnaires collected from special administrative areas (Bangkok and Pattaya) amounted to 35% of the total, reflecting their status as capitals and economic zones with dense populations. Seven percent originated from city municipalities located in only 30 provinces. The town and subdistrict municipalities each contributed 20% of the total questionnaires. Another 18% came from subdistrict administrative organization areas. Regarding residential activity, respondents were asked to select the category that best described their area. The finding revealed that respondents lived in commercial areas (49%), agricultural areas (25%), tourist areas (16%), and industrial areas (10%). In terms of education level, 63% of the respondents had a bachelor's degree, while 22% held higher degrees and 15% held lower degrees. The number of household members ranged from 1 to 11, and 65% of the respondents belonged to families with 3–5 members. Moreover, most respondents (48%) had a household income of 15,000–50,000 baht/month/household.

Short-lifespan or single-use plastics consumption was different for each kind of plastic. For example, an examination of plastic bags revealed a range of consumption rates from 0 to 25 pieces per household per week (pieces/household/week), with the majority falling within the 3–5 pieces per household per week range (48%). However, some households abstained from using plastic bags in their daily lives due to environmental concerns and the influence of the "plastic bag ban" campaign implemented by convenience stores. Respondents consumed snack bags at a rate of 1–3 pieces/household/week (49%), while 77 households did not purchase snacks due to health concerns. Conversely, a few households indulged in snacks, acquiring more than 20 pieces per week. Food boxes, foam boxes, plastic cutlery, plastic cups, condiment sachets, face/body cream sachets, detergent/laundry sachets, and plastic bubbles were most frequently used at 1–3 pieces/household/week. Many households (18–53%) reported zero consumption in these categories, indicating a low consumption rate, especially for face/body cream sachets. Regarding diapers and sanitary napkins, the highest consumption rate exceeded 20 pieces per household per week (20%), while 23% of households did not use this category due to being exclusively composed of male members. Table 2 presents the average consumption rates of single-use plastic products. The results showed that the average household consumed single-use plastics at 63 pieces/household/week or 9 pieces/household/day. On average, Thais consumed approximately three pieces of single-use plastics per capita per day (pieces/cap/day), which is lower than the 2018 government report of eight plastic bags per capita per day [7,35]. This difference might be due to the policies launched during the last few years, such as the Roadmap on Plastic Waste Management 2018–2030 [10], the ban on free plastic bags in convenience stores, and plastic waste recycling projects launched by hypermarkets as corporate social responsibility (CSR) initiatives [36]. Furthermore, higher plastic waste awareness affected consumers' behaviors. The United Nations' Sustainability Development Goals (SDGs) and the notion of the circular economy have also raised awareness of plastic pollution. Therefore, many respondents avoided using single-use plastics.

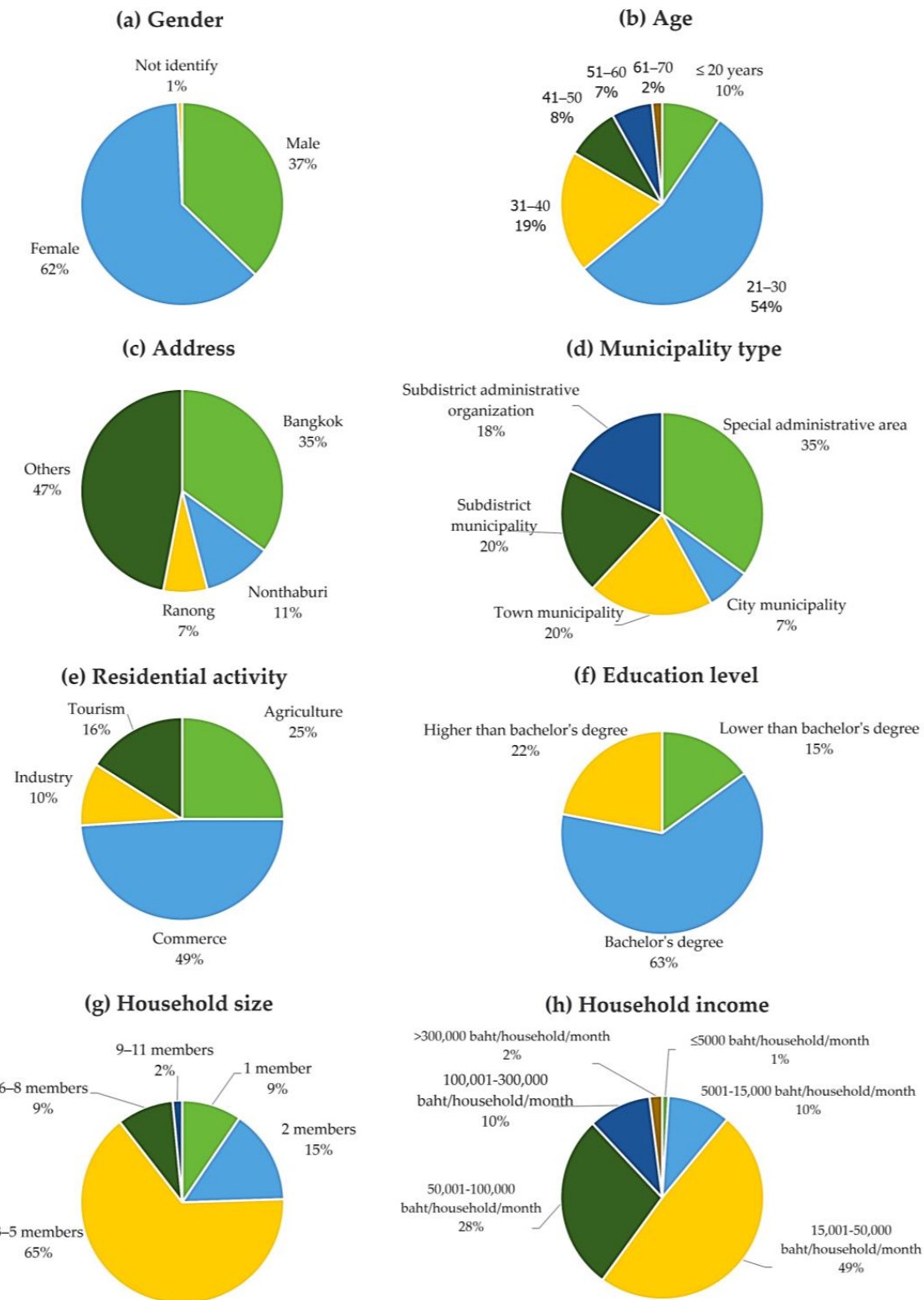

**Figure 2.** Demographic details of respondents (Total questionnaires = 567).

The masses of each category of single-use plastics were collected and measured to transform the pieced unit into a mass unit. The study found that, among the types of single-use plastics, Thais mainly consumed water bottles (17.46 g/household/day), food boxes (17.43 g/household/day), beverage bottles (15.66 g/household/day), trash bags (12.54 g/household/day), and plastic cups (11.09 g/household/day). They used other

single-use plastics at rates of less than 5 g/household/day. For example, plastic bags were consumed at about 2 g/household/day. The overall single-use plastic consumption was 91.17 g/household/day or 30.09 g/cap/day.

**Table 2.** Average consumption rates of single-use plastic products.

| Category | Average Consumption Rate | |
|:---:|:---:|:---:|
| | Piece/Household/Day | Piece/Cap/Day |
| Plastic bags | 1.16 | 0.38 |
| Food bags | 0.99 | 0.31 |
| Snack bags | 0.65 | 0.21 |
| Trash bags | 0.63 | 0.20 |
| Food boxes | 0.59 | 0.21 |
| Foam boxes | 0.37 | 0.12 |
| Plastic cutlery | 0.38 | 0.13 |
| Plastic cups | 0.55 | 0.20 |
| Straws | 0.75 | 0.25 |
| Water bottles | 1.01 | 0.33 |
| Beverage bottles | 0.66 | 0.21 |
| Condiment sachets | 0.42 | 0.13 |
| Face/body cream sachets | 0.23 | 0.07 |
| Detergent/laundry sachets | 0.31 | 0.09 |
| Plastic bubbles | 0.25 | 0.08 |
| Diapers/sanitary napkins | 0.10 | 0.04 |
| Total | 9.05 | 2.95 |

Medium-use plastics exhibited a collective consumption rate of an average of 10 pieces per household per month or 3 pieces per capita per month, as shown in Table 3. Most households used medium-use plastic products at a rate of about 1–3 pieces over a few months. For example, the use of refilled detergent/laundry bags ranged from 0 to more than 25 pieces but were mainly used at a rate of 1–3 pieces in a few months, accounting for 58% of households. On average, refilled detergent/laundry bags were utilized at a rate of two pieces/household/month (0.5 pieces/cap/month). Other categories displayed an average consumption rate of approximately one piece/household/month (0.2–0.3 piece/cap/month). When considering weight measurements, the overall consumption of medium-use plastics amounted to 579.85 g/household/month (19.33 g/household/day) or 173.38 g/cap/month (5.78 g/cap/day). Among the different categories, office supplies had the lowest consumption rate at 6.04 g/household/month, while condiment bottles had the highest rate at 151.17 g/household/month. Notably, the consumption rate of medium-use plastics was lower than that of single-use plastics due to the longer lifespan of the former, resulting in a less frequent need to repurchase.

Long-use plastics were found to have an overall consumption rate of 50 pieces/household/year or 16 pieces/cap/year, as illustrated in Table 4. Notably, each category displayed distinct variations in consumption rates. For example, reusable plastic bottles and clothing hangers were commonly utilized at 7–8 pieces/household/year. In terms of mass, the consumption of long-use plastics by Thais amounted to an average of 24.29 kg/household/year or 6.40 kg/cap/year. For instance, cabinets and beds demonstrated the highest consumption rate, approximately 5.06 kg/household/year, and tree pots exhibited the lowest rate, at about 0.16 kg/household/year. Notably, the consumption of long-use plastics was observed to be relatively low in terms of pieces/household/day. However, long-use plastics exhibited

a higher weight/household/day when compared to single-use and medium-use plastics. This can be attributed to the fact that long-use plastic products generally have a higher weight per individual item. Additionally, the infrequent repurchase of long-use plastics and the limited demand for various categories within this group were noteworthy findings. For example, 61% of respondents reported not purchasing plastic cabinets or beds within the past year.

**Table 3.** Average consumption rates of medium-use plastic products.

| Category | Average Consumption Rate | |
| --- | --- | --- |
| | Piece/Household/Month | Piece/Cap/Month |
| Refilled detergent/laundry bags | 1.65 | 0.51 |
| Detergent/laundry bottles | 0.98 | 0.28 |
| Shampoo/conditioner bottles | 1.02 | 0.32 |
| Soap bottles | 0.89 | 0.27 |
| Face/body cream bottles | 0.82 | 0.26 |
| Condiment bottles | 0.98 | 0.28 |
| Alcohol bottles/sprays | 0.69 | 0.21 |
| Jars | 0.71 | 0.22 |
| Office supplies | 0.64 | 0.20 |
| Toothbrushes | 0.83 | 0.26 |
| Plastic tubes | 0.85 | 0.27 |
| Total | 10.07 | 3.09 |

**Table 4.** Average consumption rates of long-use plastic products.

| Category | Average Consumption Rate | |
| --- | --- | --- |
| | Piece/Household/Year | Piece/Cap/Year |
| Reusable plastic bottles | 7.22 | 2.24 |
| Shoulder bags/backpacks | 3.20 | 0.98 |
| Clothes | 5.83 | 1.93 |
| Clothing hangers | 7.87 | 2.54 |
| Hanging rails | 2.37 | 0.69 |
| Shelves | 2.41 | 0.72 |
| Boxes/drawers | 2.57 | 0.80 |
| Cabinets/beds | 1.63 | 0.48 |
| Tables/chairs | 1.85 | 0.57 |
| Plates/bowls/cups/glasses | 3.88 | 1.16 |
| Kitchenware | 2.78 | 0.84 |
| Buckets | 2.78 | 0.87 |
| Toys | 2.80 | 0.89 |
| Tree pots | 2.98 | 0.93 |
| Total | 50.16 | 15.64 |

The overall plastic consumption in Thailand was determined to be 177.04 g/household/day or 53.40 g/cap/day. This means that Thais consume plastic products at an annual rate of approximately 64.62 kg/household (19.49 kg/cap), which is lower than the government's

reported figure of 40 kg/cap in 2018 [7,35]. This decline in consumption can be attributed to the implementation of plastic reduction policies, collaborative efforts within the private sector, and increased environmental awareness among the population [36]. The results also revealed that Thais consume plastic products at approximately half the world's average plastic consumption rate of 0.12 kg/cap/day [5,6]. Moreover, Thai consumption was also lower than that in North America, Europe, and Japan. This trend aligns with previous studies indicating that developed countries tend to consume more plastic than developing countries such as Thailand [7]. One possible explanation for this disparity could be that Thais often reuse plastic products and collect valuable plastic items for selling at scrap shops.

In comparison, China exhibits high plastic production and consumption rates, and its domestic plastic usage and global market share have shown consistent annual growth, particularly concerning health-related applications. In 2019, the per capita plastic consumption in China reached approximately 69 kg/year (189 g/day), with plastic waste generation reaching 47 kg/year (129 g/day), surpassing the findings of the present study for Thailand [37]. As for environmental concerns (e.g., GHG emissions, waste management, microplastic contamination), China banned most plastic waste imports in 2017 and has supported and enhanced the domestic recycling of plastic. The circular innovation of plastics has evolved rapidly. The United States also faces challenges due to its substantial plastic consumption of 255 kg per capita per year [37]. Multiple US states—as well as Portugal, India, and other countries— are in the process of banning single-use plastics. Meanwhile, the European Union is implementing a plastic packaging tax. In line with this global concern, Thailand has also launched a plastic roadmap to reduce single-use plastic and to enhance its circular plastic economy by up to 100% in 2030 [10]. Furthermore, international cooperation to tackle plastic pollution has gained momentum through the signing of various binding and non-binding agreements by many nations, including the United Nations Convention on the Law of the Seas (UNCLOS), the Basel Convention, the Clean Seas Pact, and the Plastic Waste Partnership (PWP) of the Basel Convention. These initiatives demonstrate the collective commitment to addressing the global challenge of plastic pollution and underscore the importance of international collaboration in finding sustainable solutions.

The analysis of respondents' behaviors revealed distinct patterns in the consumption of single-use, medium-use, and long-use plastics based on residential locations. Specifically, respondents residing in special administrative areas, such as Bangkok and Pattaya, demonstrated a higher tendency to consume single-use plastic products than those living in city, town, or subdistrict municipalities and subdistrict administrative organizations. This trend can be attributed to the higher population density in the former areas, which provides individuals with more opportunities to purchase food and other items packaged in single-use plastics. Conversely, the consumption of medium-use and long-use plastics did not exhibit significant relationships to municipality types. Medium-use plastics displayed similar consumption rates across all municipality types, averaging approximately 5–7 g/cap/day. Regarding long-use plastics, the subdistrict administrative organization exhibited the highest consumption rate at 20.46 g/cap/day, while the city municipality exhibited the lowest consumption rate at 12.91 g/cap/day. Notably, residents of urban areas, including special administrative areas, city municipalities, and town municipalities, consumed more single-use plastics and fewer long-use plastics compared to individuals in rural areas. In rural areas, where access to commodities requires travel to distant locations, the consumption of single-use plastics tended to be lower. However, people in rural areas preferred using long-use plastic products for their daily needs. Figure 3 illustrates the impact of municipality types on plastic consumption.

This study categorized residential activities into four distinct categories. The findings revealed a considerable consumption of single-use plastics across all residential activities, ranging from 20.99 to 37.32 g/cap/day. Medium-use plastics were predominantly utilized within commercial and industrial areas, while long-use plastics were primarily employed

in industrial and tourist areas. In contrast, the agricultural areas exhibited the lowest consumption rates for all types of plastics. This might be because people in agricultural regions have different consumption patterns and lifestyle choices.

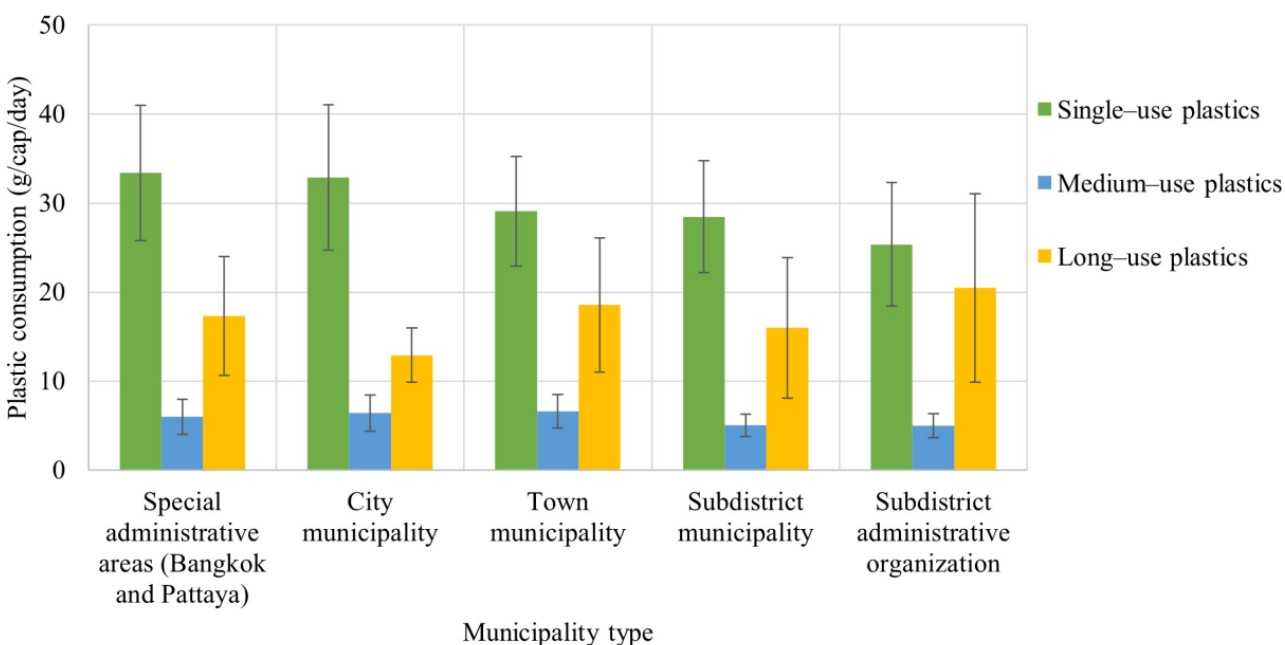

**Figure 3.** Plastic consumption by municipality type.

Concerning respondents' education levels, the findings indicated that the respondents with less education used fewer single-use plastics than those with more education, as shown in Figure 4. In addition, those with bachelor's degrees displayed a greater tendency to use medium-use and long-use plastics. These results were opposite to Jeżewska-Zychowicz and Jeznach (2015) as well as Jesevičiūtė-Ufartienė (2019), who found that people with higher educational attainment possess heightened environmental awareness and are more inclined to minimize their usage of plastic packaging [26,38]. However, this study corroborated the findings of Afroz et al. (2017) and Madigele et al. (2017), which revealed that knowledge and environmental behavior are not necessarily interconnected [39,40]. In the case of Thailand, people with more education primarily work in big cities, such as Bangkok, or in metropolitan areas, where they have more opportunities to purchase items, especially those in plastic packaging, thereby potentially contributing to the observed patterns. As mentioned earlier, the results pertaining to educational levels align with the findings regarding municipality types, whereby individuals residing in urban areas demonstrate a higher proclivity for using plastic products than their rural counterparts. Therefore, the outcomes regarding education level aligned with the results related to municipality type.

Household size played a significant role in determining the level of plastic consumption. The findings revealed a clear relationship between larger household sizes and decreased plastic consumption, as presented in Figure 5. Individuals who lived alone demonstrated the highest utilization rates across all plastic types. Specifically, single adults living alone exhibited a consumption rate of 66.36 g/cap/day for single-use plastics, surpassing that of larger families comprising a minimum of two members (13.97–33.88 g/cap/day). This result can be attributed to the fact that individuals living alone are forced to obtain their own plastic products, while larger families have the advantage of purchasing and sharing them among all household members. Furthermore, residing with family members appeared to foster greener attitudes, greater plastic separation for recycling purposes, and a heightened awareness of sustainability [23,29,39]. Notably, an increase in the number of

family members corresponded to a substantial reduction in plastic consumption, ranging from 74% to 98%.

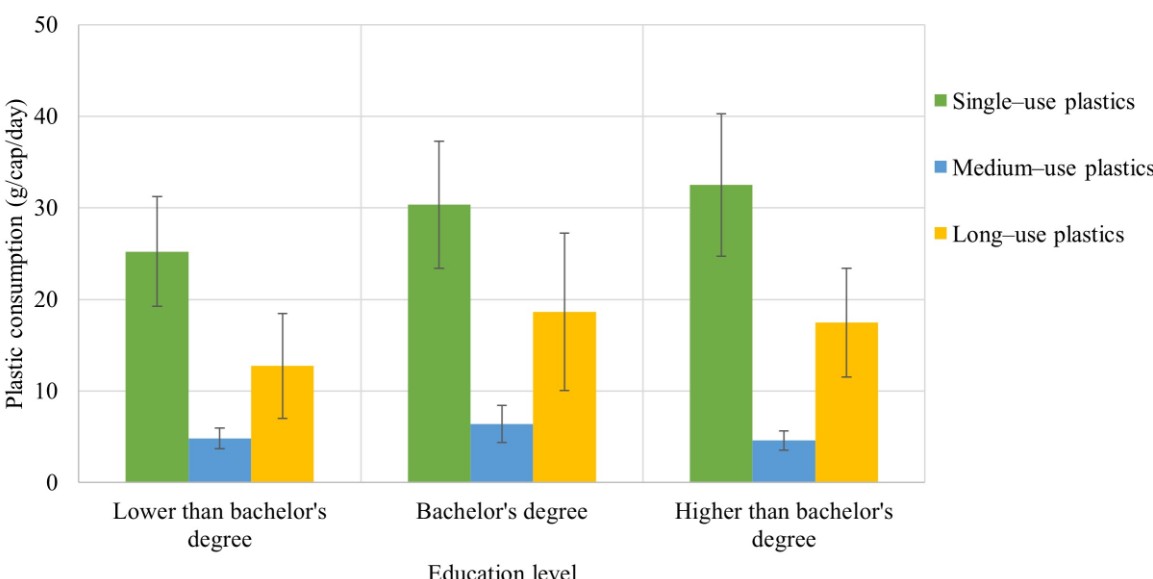

**Figure 4.** Plastic consumption by education level.

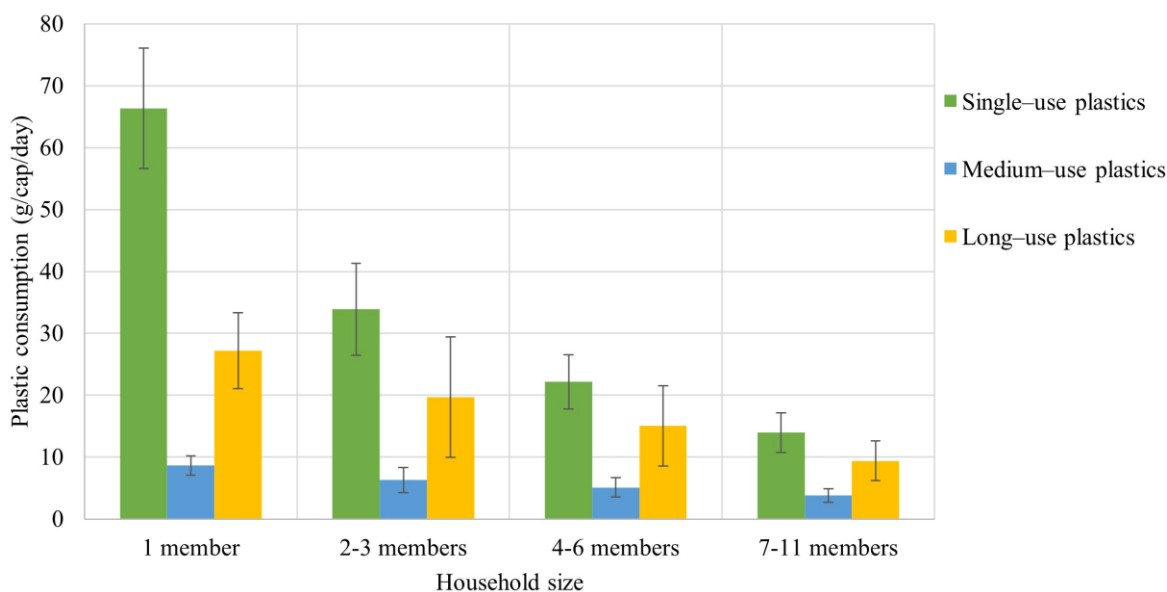

**Figure 5.** Plastic consumption by household size.

This study did not uncover discernible trends in the relationship between household income and plastic consumption. Nevertheless, the families with the lowest income (≤5000 baht/month) demonstrated the lowest levels of plastic consumption compared to other income groups. Single-use plastics were found to be consumed at high rates across almost all family income levels. Previous research has found that family income influences plastic consumption behavior, but the findings are diverse [26,29,39,40]. For instance, Jesevičiūtė-Ufartienė (2019) [26] and Afroz et al. (2017) [39] reported that people with higher incomes were less willing to pay for plastic products because they spent money more conscientiously. Conversely, Madigele et al. (2017) [40] found that people with higher incomes are willing to pay for plastic products, particularly plastic bags, for convenience, even those with a heightened environmental awareness.

Therefore, several factors significantly influenced the plastic consumption rate in Thailand, including municipality type, residential activity, education level, and household size. A clear relationship was observed between household sizes and plastic consumption, whereby an increase in family size corresponded to lower consumption levels. Plastic consumption varied noticeably based on residential activity, with agricultural areas exhibiting lower rates than other areas, such as industrial, commercial, and tourist zones, which displayed higher rates. Municipality type and education level showed obvious trends only in relation to single-use plastic consumption, while no trends were observed for medium-use and long-use plastic consumption. In addition, household income was not significantly related to plastic consumption.

### 3.2. Multiple Linear Regression Analysis

Residential activity and household size significantly affected single-use plastic consumption behaviors, with $p$-values less than 0.05. Medium-use plastic consumption was influenced by household size and household income. Long-use plastic consumption was affected by household size. Thus, household size has a significant effect on plastic consumption in Thailand. This remarkable result aligned with the behavioral results mentioned earlier. After removing the factors with a $p$-value $\geq 0.05$, an equation for each plastic type was developed, as shown in Equations (3)–(5). However, these equations could not be accepted because the R squared values were too low, at 0.19, 0.04, and 0.02 for single-use, medium-use, and long-use plastics, respectively.

$$\text{Single-use plastic (g/cap/day)} = +4.23\,\text{R} - 6.68\,\text{S} + 45.83 \quad \text{R}^2 = 0.19 \tag{3}$$

$$\text{Medium-use plastic (g/cap/day)} = -0.74\,\text{S} + (7.27 \times 10^{-6})\,\text{I} + 8.00 \quad \text{R}^2 = 0.04 \tag{4}$$

$$\text{Long-use plastic (g/cap/day)} = -2.55\,\text{S} + 26.93 \quad \text{R}^2 = 0.02 \tag{5}$$

R represents residential activity in the equations, with agriculture = 1, commerce = 2, tourist area = 3, and industrial activity = 4. S is the household size (1–11 members), and I is the household income (baht/household). Based on the equations, it should be noted that the multiple linear regression was not a suitable model for predicting plastic consumption. Furthermore, this study did not conform to alternative models, such as logarithmic and polynomial regressions.

### 3.3. Greenhouse Gas (GHG) Emissions

Greenhouse gas (GHG) emissions resulting from the end-of-life stage of household plastic consumption were quantified based on the disposal behaviors reported by the respondents. In urban areas, people discard their garbage into public waste containers, with local municipalities taking charge of collecting, treating, and disposing of waste within their districts. People in rural areas where the waste management system does not extend its reach resorted to disposing of waste through methods like dumping and covering with soil or open burning within their residential localities. At the same time, they sometimes sold valuable plastics at scrap shops, commonly found throughout the country. Most single-use plastics were discarded in public trash bins near the respondents' homes, accounting for 63–93% of cases. For instance, plastic bags were predominantly disposed of in trash bins, constituting 516 households (91% of total households). The remaining plastic bag waste was managed through burning (7.8%), selling it to scrap shops (1%), and burial (0.2%). For medium-use plastics, a significant portion was disposed of in public trash bins, similar to the trends observed for single-use plastics. Approximately 75–93% of households employed this disposal method. Specific items, such as detergent/laundry bottles, shampoo/conditioner bottles, soap bottles, face/body cream bottles, condiment bottles, alcohol bottles/sprays, and jars, were sold to scrap shops for recycling purposes, accounting for approximately 15–21% of households. A lower percentage of households (0.2–4%) sold refilled detergent/laundry bags, office supplies, toothbrushes, and plastic

tubes to scrap shops. People faced difficulty selling some of these items—particularly office supplies, toothbrushes, and plastic tubes—to scrap shops due to their complex composition and limited quantity for collection. Approximately 4–7% of households openly burned medium-use plastic products on their property, while the remaining portion was buried. Like single-use and medium-use plastics, long-use plastics were primarily disposed of in public bins, accounting for 64–81% of cases. Approximately 13–32% of households sold long-use plastics to scrap shops for recycling purposes. The remaining long-use plastics were managed through burning or landfilling within households' areas. Thus, Thais predominantly manage their plastic waste in two main ways: disposing of it in public trash bins and selling it to scrap shops.

Local governments organize public trash bins. After collection, municipal waste undergoes transfer either to landfills or incineration plants. Due to their high capital and operational costs, incineration plants are predominantly located in major cities like Bangkok and Phuket. The high waste elimination rate associated with incineration raises concerns about environmental pollution due to combustion, and the plants necessitate a pollution control system. By contrast, engineering landfills offer a more cost-effective alternative to incineration; these require leachate, landfill gas, and groundwater monitoring as environmental control measures. The recycling of plastic waste represents the most environmentally and economically viable solution. However, the collection process for plastic waste poses significant challenges. Mechanical and chemical recycling demonstrate advantages over using virgin plastics from fossil feedstock due to the latter's global warming potential and cumulative energy demand. Chemical recycling exhibits superior cost-effectiveness and carbon efficiency compared to mechanical recycling [41]. Open dumping and open burning are the worst forms of waste management, capable of contaminating the atmosphere, land, and water sources with harmful pollutants.

According to the emissions per unit in Table 1, GHG emissions due to plastic consumption were determined, as shown in Table 5. End-of-life single-use plastics emitted GHG at 78.54 $gCO_2eq$/household/day or 25.14 $gCO_2eq$/cap/day. Food boxes (5.77 $gCO_2eq$/cap/day) released the most GHGs among single-use plastic products because of the high consumption rate and high rate of disposal in public trash bins. The lowest GHG emissions came from foam boxes (0.06 $gCO_2eq$/cap/day); these are now used less for food packaging as people prefer plastic food boxes made from polypropylene and polystyrene. Medium-use plastics released GHG emissions of approximately 16.71 $gCO_2eq$/household/day or 4.81 $gCO2eq$/cap/day at the end-of-life stage. Condiment bottles emitted the most GHGs (1.16 $gCO_2eq$/cap/day), while office supplies released low GHG emissions at 0.07 $gCO_2eq$/cap/day. Long-use plastics had GHG emissions at 53.96 $gCO_2eq$/household/day or 13.73 $gCO_2eq$/cap/day. Due to their heavy weights, cabinets, and beds showed the highest GHG emissions among long-use plastics at about 3.29 $gCO_2eq$/cap/day. Tree pots had the lowest GHG emissions at 0.11 $gCO_2eq$/cap/day due to their low consumption rate. Therefore, the overall GHG emissions for all three types of plastics were 149.21 $gCO_2eq$/household/day (43.68 $gCO_2eq$/cap/day). Based on the population in Thailand (66,090,475 people in 2022 [30]), Thailand's GHG emissions due to end-of-life plastics were estimated at 2887.04 $tonCO_2eq$/day or 1.05 million $tonCO_2eq$/year. End-of-life plastic waste GHG emissions differ by country depending on the waste management system. For example, end-of-life plastic films, trash bags, bottles, detergent bottles, cutlery, and others released 20–25% of the total 470.1 $kgCO_2eq$ emissions per year throughout their life cycle [42]. Paolo et al. (2022) reported that waste management includes three different scenarios: incineration, landfill, and recycling [42]. Luan et al. (2023) studied the different kinds of plastics (e.g., polypropylene, polystyrene, polyethylene) to estimate life cycle emissions [43]. They reported that in 2020, China's GHG emissions from end-of-life plastics included 9.9 million $tonCO_2eq$ from landfills, 107.0 million $tonCO_2eq$ from incineration, and 0.1 million $tonCO_2eq$ from untreated management. Moreover, global end-of-life emissions are predicted to increase continuously from 200 million $tonCO_2eq$/year in 2019 to 500 million $tonCO_2eq$/year in 2060 [17,37]. Thus, even though Thailand's GHG

emissions from plastic products amounted to only around 0.5% of global emissions, high plastic consumption should be controlled, along with plastic production and plastic waste management, to achieve net zero emissions in the future.

**Table 5.** Greenhouse gas emissions from the end-of-life stage of household plastic consumption.

| Plastic Type | Average Greenhouse Gas Emissions | |
|---|---|---|
| | gCO$_2$eq/Household/Day | gCO$_2$eq/Cap/Day |
| Single-use plastics | 78.54 | 25.14 |
| Medium-use plastics | 16.71 | 4.81 |
| Long-use plastics | 53.96 | 13.73 |
| Total | 149.21 | 43.68 |

Future solutions to address plastic pollution encompass various strategies, including using biodegradable plastics and implementing circular economy initiatives, modifications in consumer behaviors, and supportive regulatory measures. Bioplastics, also known as biopolymers, are a class of polymers derived from renewable sources, such as plants, agricultural waste, or microorganisms. Biodegradable plastics significantly reduce degradation time, but their practical implementation in real-world contexts remains challenging [37]. Continued research and innovation in the field of bioplastics aim to improve their performance and cost-effectiveness, further promoting their adoption as a sustainable solution to reduce plastic waste. Bioplastics have various applications, such as hydrogel and agricultural nutrient delivery [44,45]. The circular economy is expected to play an important role in plastic production, which will be reflected in waste management. Redesigning plastic products can facilitate more efficient recycling processes, leading to increased plastic recycling rates. Manufacturers should consider product design in a manner that minimally disrupts existing production processes. Consumers also play a crucial role in terms of plastic waste separation and collection. Additionally, government intervention through policies and regulations, such as bans on single-use plastics, incentivized design practices, and taxes on non-recycled waste generation, is crucial in supporting the private sector and citizens. A collective and simultaneous approach to implementing these solutions is necessary. These strategies can lead to a decrease in GHG emissions over the long term. However, industries tend to use $CO_2$ capture innovation to reduce $CO_2$ rapidly from the combustion process [46–48]. The sorbents typically used include zeolites, carbon-based materials, and aluminum formate (Al(HCOO)$_3$) [49]. These sorbents not only effectively adsorb $CO_2$ emissions but also demonstrate a remarkable capacity for selective $CO_2$ capture with heightened efficiency [49,50]. Therefore, commitment from all sectors to address GHG emissions could contribute significantly to reducing overall global emissions.

## 4. Conclusions

Plastic consumption has become a significant global concern due to its environmental impact. This study revealed that single-use plastics are Thailand's most commonly consumed type of plastic. The overall single-use plastic consumption was estimated to be 30.09 g/cap/day. Medium-use plastics, which have a longer lifespan than single-use plastics, exhibited an average consumption rate of three pieces/cap/month. Long-use plastics, characterized by their durability and infrequent repurchase, had an overall consumption rate of 16 pieces/cap/year. The environmental impact was measured in terms of GHG emissions in light of the climate change crisis. The emissions were calculated based on the end-of-life stage for each plastic category. Overall, the GHG emissions from single-use, medium-use, and long-use plastics amounted to 43.68 g/cap/day. Based on Thailand's population, the country was estimated to emit around 2887.04 tonCO$_2$eq/day or 1.05 million tons/year due to plastic waste. These emissions contribute to global GHG emissions, highlighting the need to control plastic consumption and improve waste man-

agement practices. The consequences of plastic pollution pose potential risks to human health and threaten ecosystems, thereby jeopardizing biodiversity, disrupting food chains, and adversely affecting marine life. Urgent action in the form of individual behavior modification is crucial. This entails reducing plastic consumption, reusing plastic products, and adopting practices for the proper separation and disposal of plastic waste to dedicated scrap shops, all of which could significantly mitigate the environmental impacts of plastic pollution. While the complete eradication of plastic usage may be unfeasible, initiating these behavioral changes constitutes an important step in ensuring the well-being of humanity and supporting sustainability for future generations.

**Author Contributions:** Conceptualization, V.K., W.K. and P.T.; methodology, V.K.; investigation, V.K.; writing—original draft preparation, V.K.; writing—review and editing, W.K. and P.T. All authors have read and agreed to the published version of the manuscript.

**Funding:** This research was supported by the University Income Fund, Naresuan University, grant number R2566C002.

**Institutional Review Board Statement:** Not applicable.

**Informed Consent Statement:** Not applicable.

**Data Availability Statement:** Not applicable.

**Acknowledgments:** This work was funded by Naresuan University, namely, the University Income Fund (Grant no. R2566C002). The authors thank all respondents in Thailand for answering the questionnaires. Moreover, this study was supported by the Hub of Talent, the Center of Excellence on Hazardous Substance Management, and the National Research Council of Thailand (NRCT), Thailand.

**Conflicts of Interest:** The authors declare no conflict of interest.

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
