# Peer review of "CO2 Emissions from Plastic Consumption Behaviors in Thailand"

_sustainability, doi:10.3390/su151612135_

Round 1

Reviewer 1 Report

The manuscript titled "CO2 emissions from plastic consumption behaviours in Thailand" discusses an intriguing topic: the CO2 emissions caused by plastic products. The authors have systematically studied 567 independent people of different ages, genders, and education levels. This study provides interesting insights into the consequences of plastic consumption in Thailand. The manuscript is well-written and concise. However, the authors should address the following points before submitting the manuscript for publication:

  1. The authors should provide a detailed introduction about the consequences of plastic, particularly CO2 emissions, which cause global warming.
  2. The authors should discuss how Thailand is rapidly experiencing global warming and CO2 emissions, and plastics. They should provide data on Thailand's CO2 emissions and information on how these emissions contribute to climate change. 
  3. Although this manuscript focuses on plastic consumption in Thailand, the authors should also discuss how other countries, particularly major plastic-producing countries, face the same consequences. This comparative analysis will help to highlight the global nature of the plastic pollution problem and the need for international cooperation to address it.
  4. The authors should discuss future directions for alternatives to plastic and what precautions should be taken.
  5. The authors should cite recent literature that addresses CO2 emissions and the materials that are effective in addressing this issue. The following papers are suggested: Journal of the American Chemical Society 145 (17), 9850-9856, Science Advances 8 (44), eade1473 and  J. Am. Chem. Soc. 2023, 145, 21, 11643–11649
  6. The conclusion should discuss the future consequences of plastic pollution and the severe problems that we need to face if we continue to rely on plastic. The authors should emphasize the need for urgent action to reduce plastic production and consumption.

A minor English editing required 

Author Response

We appreciate the time and effort that the reviewer dedicated to providing feedback on our manuscript and are grateful for the insightful comments on and valuable improvements to our paper. We have revised and tried our best to respond to every recommendation, as kindly find below or please see an attachment.

Reviewer 1:

  1. The authors should provide a detailed introduction about the consequences of plastic, particularly CO2 emissions, which cause global warming.

                Thank you very much for your kind suggestions. As suggested by the reviewer, we wrote a new paragraph with more detail about consequences of CO2 emissions on page 2, line 79-97 that “Nowadays, international agreement advocates for the limitation of the global average temperature increase to 1.5-2°C, which is caused by GHG emissions [15]. The cumulative CO2 emission should not exceed 1,000 GtCO2eq during the period from 2000 to 2050. However, quantity of 234 GtCO2eq was released and accrued in the atmosphere between 2000 and 2006 [16]. The Organization for Economic Cooperation and Development (OECD) reported that the significant role of plastic in contributing to GHG emissions, with the complete life cycle of plastic products responsible for emitting 1.8 gigatons of carbon dioxide, accounting for 3.7% of global emissions in 2019 [17]. It tends to increase to 4.3 gigatons of carbon dioxide (4.5% of global emissions) in 2060 that could be accounted for more than 19% of the total carbon budget in the world [18]. Thus, plastic is one of the big sources of GHG emissions, contributing to the escalation of global temperatures. The consequences of surpassing the 1.5-2°C warming threshold are manifold, encompassing a range of risks to both natural and human systems, including sea-level rising, ecosystem losing, land-use changing, food security losing, heatwave facing, and so on [19]. Many cases showed the adaptative response of the body size of creature to survive in the climate change [20]. For example, the alpine (V. biflora) is sensitive to climate change by changing the leaf size and number of leaves [21]. Notably, if global warming were to exceed 4°C, the alpine species could face a mortality rate ranging from 90% to 100%, as found in the study by Cui et al. (2018).”

  1. The authors should discuss how Thailand is rapidly experiencing global warming and CO2 emissions, and plastics. They should provide data on Thailand's CO2 emissions and information on how these emissions contribute to climate change.

                Thank you for the comment. As suggested by the reviewer, we wrote a new paragraph of CO2 emission in Thailand and the consequences of it on page 2-3, line 98-108 that “Thailand is an industrialized and rapidly developing country, which has led to an increase in greenhouse gas emissions. The primary contributors to GHG emissions in Thailand encompass the energy sector, agriculture, industry, and waste management [14]. Over the past two decades, Thailand's total GHG emissions have experienced a substantial increase, rising from 246 MtCO2eq in 2000 to exceeding 350 MtCO2eq presently [14]. The emissions have increased Thailand’s annual mean maximum temperature. Thailand has faced the occurrence of severe weather events such as storms, floods, and droughts that resulted in economic damages. An example of this was the 2011 flood, which led to the declaration of 65 out of 77 provinces as disaster areas. This catastrophic event resulted in extensive damage to agricultural, industrial, and residential sectors, with an estimated cost of approximately 45.7 billion US dollars [22].”

  1. Although this manuscript focuses on plastic consumption in Thailand, the authors should also discuss how other countries, particularly major plastic-producing countries, face the same consequences. This comparative analysis will help to highlight the global nature of the plastic pollution problem and the need for international cooperation to address it.

                Thank you for pointing this out. We revised and discussed more about the plastic consumption and the action to decrease plastic pollution on page 10, line 317-337 that “For other countries, China exhibits high plastic production and consumption rates, and its domestic plastic usage and global market share have shown consistent annual growth, particularly concerning health-related applications. In 2019, the per capita plastic consumption in China reached approximately 69 kg/year (189 g/d), with plastic waste generation reaching 47 kg/year (129 g/d), surpassing the findings of the present study [36]. As for environmental concerns (e.g., GHG emission, waste management, microplastic contamination), China has banned most imports of plastic waste since 2017. The domestic recycling of plastic has been supported and enhanced in the country. The circular innovation of plastics has evolved rapidly. The United States also faces challenges due to its substantial plastic consumption, which stands at 255 kg per capita per year [36]. Multiple states in the United States are proceeding to ban single-use plastics as well as Portugal, India, and so on. While the European Union is proceeding on plastic packaging tax. In line with this global concern, Thailand has also launched a plastic roadmap to reduce single-use plastic and to enhance a circular plastic up to 100% in 2030 [10]. Furthermore, international cooperation in tackling plastic pollution has gained momentum through the signing of various binding and non-binding agreements by many nations, e.g., United Nations Convention on the Law of the Seas (UNCLOS), the Basel Convention, the Clean Seas Pact, and the Plastic Waste Partnership (PWP) of the Basel Convention. These initiatives demonstrate the collective commitment to address the global challenge of plastic pollution and underscore the importance of international collaboration in finding sustainable solutions.”

  1. The authors should discuss future directions for alternatives to plastic and what precautions should be taken.

                We agree with the reviewer’s suggestion. We added the discussion about future directions of the plastics on page 15, line 525-543 that “Future solutions to address plastic pollution encompass various strategies, including the implementation of biodegradable plastics, circular economy initiatives, modifications in consumer behaviors, and supportive regulatory measures. Bioplastics, also known as biopolymers, are a class of polymers derived from renewable sources, such as plants, agricultural waste, or microorganisms. Biodegradable plastics are a great idea in reducing the degradation time, but practical implementation in real-world environments remains challenging [36]. Continued research and innovation in the field of bioplastics aim to improve their performance and cost-effectiveness, further promoting their adoption as a sustainable solution to reduce plastic waste. Bioplastics can be applied to various applications such as hydrogel, nutrient delivery for agricultural aspects [43-44]. The circular economy will play an important role for plastic production reflecting to waste management. The redesign of plastic products can facilitate more efficient recycling processes, leading to increased plastic recycling rates. Manufacturers should consider product design in a manner that minimally disrupts existing production processes. Consumers are also a key role in case of plastic waste separation and collection. Additionally, government intervention through policies and regulations, such as bans on single-use plastics, incentivized design practices, and taxes on non-recycled waste generation, is crucial in supporting the private sector and citizens. A collective and simultaneous approach to implementing these solutions is necessary.”

  1. The authors should cite recent literature that addresses CO2 emissions and the materials that are effective in addressing this issue. The following papers are suggested: Journal of the American Chemical Society 145 (17), 9850-9856, Science Advances 8 (44), eade1473 and J. Am. Chem. Soc. 2023, 145, 21, 11643–11649

                Thank you for your suggestion. We addressed an innovative CO2 capture with your suggested literature on page 15, line 543-548 that “These strategies can lead to a decrease in GHG emissions in the long term. However, CO2 capture innovation tends to be used to reduce CO2 rapidly from combustion process in the industries. The sorbents are typically used including zeolites, carbon–based materials, and aluminum formate (Al(HCOO)3) [45]. Therefore, commitment from all sectors towards addressing GHG emissions could contribute significantly to reducing overall global emissions.”

  1. The conclusion should discuss the future consequences of plastic pollution and the severe problems that we need to face if we continue to rely on plastic. The authors should emphasize the need for urgent action to reduce plastic production and consumption.

                Thank you for pointing this out. We revised the conclusion on page 16, line 563-571 that “The consequences of plastic pollution pose potential risks to human health and threaten ecosystems, thereby jeopardizing biodiversity, disrupting food chains, and adversely affecting marine life. Urgent action in the form of behavior modification among individuals is crucial. This entails reducing plastic consumption, reusing plastic products, and adopting practices for the proper separation and disposal of plastic waste to dedicated scrap shops, all of which could significantly mitigate the environmental impacts of plastic pollution. While a complete eradication of plastic usage may be unfeasible, initiating these behavioral changes constitutes an important step in ensuring the well-being of humanity and supporting a sustainability for future generations.”

Reviewer 2 Report

Article titled as "CO2 emissions from plastic consumption behaviors in Thailand" discussed that Plastic waste is an environmental crisis that is becoming increasingly well-documented. 7 The rapid expansion of plastic manufacturing and consumption has led to a harmful cycle of pollu- 8 tion and greenhouse gas emissions due to petroleum-based production and plastic waste disposal. 9 Plastic production and disposal depend on the consumption behavior of people. This study aimed 10 to examine the plastic consumption behavior in Thailand and its impact on climate change at the 11 end-of-life. The general information, plastic consumption, and plastic waste management were col- 12 lected by questionnaires for each product lifetime including single-use, medium-use, and long-use 13 plastics. Based on 567 questionnaires, the results showed that people consumed single-use plastic, 14 e.g., plastic bag, food container, cutlery, straws, and bottles about 9 pieces/household/d or 3 15 pieces/cap/d. The medium-use and long-use plastic were 10 pieces/household/month and 50 16 pieces/household/year, respectively. It should be remarked that population density, education, and 17 number of household members affected plastic consumption behavior, especially for single-use 18 plastic. At the end-of-life, these led to Thai people average contributing 0.15 kgCO2eq/household/d 19 of climate change. Many households have mismanaged waste by open dumping and open burning. 20 Therefore, the proper waste management will help Thailand on the path to carbon neutrality in the 21 future.

Comments:

The following points may be referred for further improvements

Authors might add a small paragraph on bioplastics and biobased polymers. Following studies might be cited for reference (https://doi.org/10.1007/s10924-023-02859-1, https://doi.org/10.1007/s10924-023-02893-z)

Figure quality should be improved

Add error bars in Figure 2, 3 & 4

May like to add a small paragraph on techno-economic analysis data as well

 Minor editing of English language required

Author Response

We appreciate the time and effort that the reviewer dedicated to providing feedback on our manuscript and are grateful for the insightful comments on and valuable improvements to our paper. We have revised and tried our best to respond to every recommendation, as kindly find below or please see an attachment.

Reviewer 2:

  1. Authors might add a small paragraph on bioplastics and biobased polymers. Following studies might be cited for reference (https://doi.org/10.1007/s10924-023-02859-1, https://doi.org/10.1007/s10924-023-02893-z)

                Thank you very much for your kind suggestions. We have addressed these comments by adding bioplastic/biopolymer on page 15, line 527-534 that “Bioplastics, also known as biopolymers, are a class of polymers derived from renewable sources, such as plants, agricultural waste, or microorganisms. Biodegradable plastics are a great idea in reducing the degradation time, but practical implementation in real-world environments remains challenging [36]. Continued research and innovation in the field of bioplastics aim to improve their performance and cost-effectiveness, further promoting their adoption as a sustainable solution to reduce plastic waste. Bioplastics can be applied to various applications such as hydrogel, nutrient delivery for agricultural aspects [43-44].”

  1. Figure quality should be improved

                Thank you for pointing this out. We changed all Figures for better quality as shown in page 4, 7, 11, and 12.

  1. Add error bars in Figure 2, 3 & 4

                Thank you for the comment. We added the error bars in Figure 3, 4 and 5 as shown in page 11 and 12.

  1. May like to add a small paragraph on techno-economic analysis data as well

       As suggested by the reviewer, we added the techno-economic analysis of plastic waste management in Thailand on page 14, line 475-490 that “The public trash bins are organized by local government. After waste collection, municipal waste undergoes transfer to either landfills or incineration plants. Incineration plants are predominantly situated in major cities like Bangkok and Phuket due to their high capital and operational costs. The high waste elimination rate associated with incineration raises environmental pollution concerns from combustion. The plants need to have a pollution control system. On the other hand, engineering landfills offer a more cost-effective alternative to incineration. The leachate, landfill gas, and groundwater monitoring are important for environmental control measures. The recycling of plastic waste represents the most environmentally and economically viable solution. Nonetheless, the collection process for plastic waste poses significant challenges. Mechanical and chemical recycling demonstrate advantages over virgin plastics from fossil feedstock based on global warming potential and cumulative energy demand. Chemical recycling exhibits superior cost-effectiveness and carbon efficiency compared to mechanical recycling [40]. In the case of open-dumping and open-burning are the worst forms of waste management, capable of contaminating the atmosphere, land, and water sources with harmful pollutants.”

Reviewer 3 Report

This paper investigates and examines plastic consumption behaviour and its impact on climate change in Thailand.The article is well-structured, with a correct conclusion, and needs the following refinements:

1.The reasons for the coefficient of 0 for Recycle (sell to junk shop) in Table 1 are suggested to be shown in the table.

2.Please verify the use of "Femail" in table 2.

3.Table 2 contains a large variety of data, indicating the distribution of statistical data, you can consider using a pie chart will be more intuitive.

4.It is recommended that the formatting colors of figure 2 be harmonized with the other bar charts.

5.The concept of GHG emission factors in article 2.4 seems redundant, why not just call them "emissions per unit"?

6.Initial capitalization on line 387.

7.The result of the analysis in 3.2 is that the multivariate linear model is not applicable to analyze this study? So what is the significance of this section?

Author Response

We appreciate the time and effort that the reviewer dedicated to providing feedback on our manuscript and are grateful for the insightful comments on and valuable improvements to our paper. We have revised and tried our best to respond to every recommendation, as kindly find below or please see an attachment.

Reviewer 3:

  1. The reasons for the coefficient of 0 for Recycle (sell to junk shop) in Table 1 are suggested to be shown in the table.

Thank you very much for your kind suggestions. We revised and explained more about zero for recycling process (sell to junk shop or scrap shop) on page 5, line 196-203 that “The recycling process was assumed to be negligible because the waste could be repurposed as raw material in various industries [31]. Preceding their sale at junk or scrap shops, plastic wastes were initially segregated at the household level. Subsequently, the scrap shops have a responsibility of collecting and preparing these materials to transfer to the recycling manufactures. In the context of this study, the emissions of recycling were identified from the household separation to the scrap shop gate, with the emission considered as zero due to the exclusion of transportation-related impacts from the analysis.”

  1. Please verify the use of "Femail" in table 2.

                Thank you very much for pointing this out. We apologize for this mistake. The “Femail” was changed to be “Female” on page 7 in Figure 2 that we changed Table 2 to Figure 2.

  1. Table 2 contains a large variety of data, indicating the distribution of statistical data, you can consider using a pie chart will be more intuitive.

                Thank you for the suggestion, we decided to change the data in Table 2 to Figure 2 on page 7 that can be more understood clearly.

  1. It is recommended that the formatting colors of figure 2 be harmonized with the other bar charts.

                Thank you very much for pointing this out. We changed Figure 3, 4, and 5 to be the same format in color on page 11 and 12.

  1. The concept of GHG emission factors in article 2.4 seems redundant, why not just call them "emissions per unit"?

We respectfully appreciated the reviewer’s comment. Emission factor is the general term of GHG emission per unit. As you mentioned, we decided to change “emission factor” to “emissions per unit” throughout the manuscript to avoid redundancy.

  1. Initial capitalization on line 387.

                Thank you very much for pointing this out. We apologize for this mistake. We corrected on page 13, line 425-426 that “Single-use plastics found that residential activities and household sizes affected plastic consumption behaviors significantly with P-values less than 0.05.”

  1. The result of the analysis in 3.2 is that the multivariate linear model is not applicable to analyze this study? So what is the significance of this section?

                Thank you for the comment. We would like to explain that the multilinear regression model is typically used to analyze waste behavior. Unfortunately, our study did not fit it. This section will enlighten that the multilinear regression cannot be applied for some cases like our study. Moreover, we tried to analyze by other models, but the results did not fit well. Thus, we added a sentence to clarify it on page 13, line 441-442 that “Furthermore, this study did not conform to alternative models, such as logarithmic and polynomial regressions.”

Reviewer 4 Report

This study aimed to examine the plastic consumption patterns of individuals in Thailand using questionnaires. The research involved gathering and analyzing demographic information, including the type of municipality, activity level in the residential area, educational background, household size, and household income. These factors were investigated to determine their correlation with the amount of consumed plastic.

The methodology applied in the manuscript is appropriate and well-described. However, from lines 134 to 146, the information is redundant compared to Figure 1.  The authors could synthesize the text.

Lines 176 to 181 could be considered part of the discussion section.

In the results section, the authors could summarize the information presented before the Tables since the information is redundant. 

In line 338, why do the Authors refer to a P-value<0.5? Did you mean 0.05? Please check it.

In line 392: "When the factors with P-value≥0.05 were cut off, the equations of all plastic types were shown in Equations 3, 4, and 5"; you must check this section. All the Equations have a Pearson coefficient that is insufficient for linear correlation.  If multiple linear regression was not a suitable method for prediction, which method might be effective? Why didn't you explore another way to correlate the data?

The article falls short in the statistical data analysis and could be improved to provide more traceable and understandable results. The correlation study mentioned in the introduction must be presented in the results.

The manuscript is written correctly and well-analyzed, in general.

Author Response

We appreciate the time and effort that the reviewer dedicated to providing feedback on our manuscript and are grateful for the insightful comments on and valuable improvements to our paper. We have revised and tried our best to respond to every recommendation, as kindly find below or please see an attachment.

Reviewer 4:

  1. The methodology applied in the manuscript is appropriate and well-described. However, from lines 134 to 146, the information is redundant compared to Figure 1.  The authors could synthesize the text.

Thank you very much for your kind suggestions. We agree with the reviewer’s suggestion that these sentences were too long and redundant. We revised these sentences specifically on page 4-5, line 155-159 that “Single-use plastics were studied, e.g., plastic carrier bags, food bags, and straws. Medium-lifespan plastics, such as detergent bottles, shampoo bottles, and toothbrushes showed longer lifespan than single-use plastics. Additionally, long-lifespan plastics were examined, such as reusable bottles, bags, and durable household items like shelves, tables, and kitchenware.”

  1. Lines 176 to 181 could be considered part of the discussion section.

                Thank you for the suggestion. We would like to maintain these sentences in the method section; however we improved and added the waste management information in Thailand in the discussion part as well on page 13, line 446-452 that “In urban areas, people discarded the garbage into public waste containers, with local municipalities taking charge of collecting, treating, and disposing of waste within their respective districts. People in rural areas where the waste management system does not extend its reach resort to disposing of waste through methods like dumping with soil covering or open burning within their residential localities. At the same time, valuable plastics are sometimes sold at scrap shops, commonly found throughout the country.”

  1. In the results section, the authors could summarize the information presented before the Tables since the information is redundant.

                We respectfully appreciated the reviewer’s comment. To avoid redundancy, we revised and cut off many sentences to shorten in the result section, especially for the results of single- and long-lifespan plastics on page 7-8, line 233-249 that “Short-lifespan plastics or single-use plastics consumptions were different for each kind of plastic. For example, an examination of plastic carrier bags revealed a range of consumption rates from 0 to 25 pieces per household per week (pieces/household/week), with the majority falling within the 3-5 pieces per household per week range (48%). However, some households abstained from using plastic carrier bags in their daily lives due to environmental concerns and the influence of the "plastic bag ban" campaign implemented by convenience stores. Snack bags were primarily consumed at a rate of 1-3 pieces/household/week (49%), while 77 households did not purchase snacks due to health concerns. Conversely, a few households indulged in snacks, acquiring more than 20 pieces per week. Food boxes, foam boxes, plastic cutlery, plastic cups, condiment sachets, face/body cream sachets, detergent/laundry sachets, and plastic bubbles were most frequently used at 1-3 pieces/household/week. In these categories, a lot of households reported zero consumption accounting for 18-53%. It indicated that the consumption rate of these categories was low, especially for face/body cream sachets. Regarding diapers and sanitary napkins, the highest consumption rate exceeded 20 pieces per household per week (20%), while 23% of households did not use this category due to being exclusively composed of male members.”

and page 9, line 287-290 that “Long-use plastics were found to have an overall consumption rate of 50 pieces/household/year, or 16 pieces/cap/year as illustrated in Table 4. Notably, each category displayed distinct variations in consumption rates. For example, reusable plastic bottles and cloth hangers were commonly utilized at 7-8 pieces/household/year.”

  1. In line 388, why do the Authors refer to a P-value<0.5? Did you mean 0.05? Please check it.

                Thank you for pointing this out. We apologize for this mistake. We changed a P-value in this context from “<0.5” to “<0.05” on page 13, line 425-426 that “Single-use plastics found that residential activities and household sizes affected plastic consumption behaviors significantly with P-values less than 0.05.”

  1. In line 392: "When the factors with P-value≥0.05 were cut off, the equations of all plastic types were shown in Equations 3, 4, and 5"; you must check this section. All the Equations have a Pearson coefficient that is insufficient for linear correlation.  If multiple linear regression was not a suitable method for prediction, which method might be effective? Why didn't you explore another way to correlate the data?

                Thank you for the comment. We would like to explain that the multilinear regression model is typically used to analyze waste behavior. In this study, we considered five factors, i.e., municipality type, residential activity, education level, household member, and household income toward the single-, medium-, and long-use plastic consumptions. Each plastic type was determined which of five factors affecting the consumption rate. Unfortunately, our study did not fit it. However, this section will enlighten that multilinear regression cannot be applied for some cases like our study. Moreover, we tried to analyze by other models, but the results did not fit well. Thus, we added a sentence to clarify it on page 13, line 441-442 that “Furthermore, this study did not conform to alternative models, such as logarithmic and polynomial regressions.”

  1. The article falls short in the statistical data analysis and could be improved to provide more traceable and understandable results. The correlation study mentioned in the introduction must be presented in the results.

       Thank you for your suggestion. We revised and discussed more on page 10, line 309-337 that “The results revealed that Thais consumed plastic products, which was approximately half the world average plastic consumption of 0.12 kg/cap/d [5-6]. Moreover, Thai consumption was also lower than that of North America, Europe, and Japan. This trend aligns with previous studies indicating that developed countries tend to have higher plastic consumption compared to developing countries like Thailand [7]. One possible explanation for this disparity could be that Thais tend to reuse plastic products and collect valuable plastic items for selling at scrap shops.

For other countries, China exhibits high plastic production and consumption rates, and its domestic plastic usage and global market share have shown consistent annual growth, particularly concerning health-related applications. In 2019, the per capita plastic consumption in China reached approximately 69 kg/year (189 g/d), with plastic waste generation reaching 47 kg/year (129 g/d), surpassing the findings of the present study [36]. As for environmental concerns (e.g., GHG emission, waste management, microplastic contamination), China has banned most imports of plastic waste since 2017. The domestic recycling of plastic has been supported and enhanced in the country. The circular innovation of plastics has evolved rapidly. The United States also faces challenges due to its substantial plastic consumption, which stands at 255 kg per capita per year [36]. Multiple states in the United States are proceeding to ban single-use plastics as well as Portugal, India, and so on. While the European Union is proceeding on plastic packaging tax. In line with this global concern, Thailand has also launched a plastic roadmap to reduce single-use plastic and to enhance a circular plastic up to 100% in 2030 [10]. Furthermore, international cooperation in tackling plastic pollution has gained momentum through the signing of various binding and non-binding agreements by many nations, e.g., United Nations Convention on the Law of the Seas (UNCLOS), the Basel Convention, the Clean Seas Pact, and the Plastic Waste Partnership (PWP) of the Basel Convention. These initiatives demonstrate the collective commitment to address the global challenge of plastic pollution and underscore the importance of international collaboration in finding sustainable solutions.”

page 14, line 475-490 that “The public trash bins are organized by local government. After waste collection, municipal waste undergoes transfer to either landfills or incineration plants. Incineration plants are predominantly situated in major cities like Bangkok and Phuket due to their high capital and operational costs. The high waste elimination rate associated with incineration raises environmental pollution concerns from combustion. The plants need to have a pollution control system. On the other hand, engineering landfills offer a more cost-effective alternative to incineration. The leachate, landfill gas, and groundwater monitoring are important for environmental control measures. The recycling of plastic waste represents the most environmentally and economically viable solution. Nonetheless, the collection process for plastic waste poses significant challenges. Mechanical and chemical recycling demonstrate advantages over virgin plastics from fossil feedstock based on global warming potential and cumulative energy demand. Chemical recycling exhibits superior cost-effectiveness and carbon efficiency compared to mechanical recycling [40]. In the case of open-dumping and open-burning are the worst forms of waste management, capable of contaminating the atmosphere, land, and water sources with harmful pollutants.”

and page 15, line 525-548 that “Future solutions to address plastic pollution encompass various strategies, including the implementation of biodegradable plastics, circular economy initiatives, modifications in consumer behaviors, and supportive regulatory measures. Bioplastics, also known as biopolymers, are a class of polymers derived from renewable sources, such as plants, agricultural waste, or microorganisms. Biodegradable plastics are a great idea in reducing the degradation time, but practical implementation in real-world environments remains challenging [36]. Continued research and innovation in the field of bioplastics aim to improve their performance and cost-effectiveness, further promoting their adoption as a sustainable solution to reduce plastic waste. Bioplastics can be applied to various applications such as hydrogel, nutrient delivery for agricultural aspects [43-44]. The circular economy will play an important role for plastic production reflecting to waste management. The redesign of plastic products can facilitate more efficient recycling processes, leading to increased plastic recycling rates. Manufacturers should consider product design in a manner that minimally disrupts existing production processes. Consumers are also a key role in case of plastic waste separation and collection. Additionally, government intervention through policies and regulations, such as bans on single-use plastics, incentivized design practices, and taxes on non-recycled waste generation, is crucial in supporting the private sector and citizens. A collective and simultaneous approach to implementing these solutions is necessary. These strategies can lead to a decrease in GHG emissions in the long term. However, CO2 capture innovation tends to be used to reduce CO2 rapidly from combustion process in the industries. The sorbents are typically used including zeolites, carbon–based materials, and aluminum formate (Al(HCOO)3) [45]. Therefore, commitment from all sectors towards addressing GHG emissions could contribute significantly to reducing overall global emissions.”

Round 2

Reviewer 1 Report

The authors overlooked several major suggestions and recommended references. It is crucial to cite and acknowledge the relevant literature in this manuscript properly. The current content does not adequately reflect the existing text. I strongly recommend incorporating citations to acknowledge the work of others in this important field. For the manuscript to be considered for acceptance, it must include appropriate citations to these referenced papers.

Author Response

          Thank you very much for your feedback on our manuscript. As suggested by the reviewer, we revised and included more relevant citations on page 15, line 544-548 that “However, CO2 capture innovation tends to be used to reduce CO2 rapidly from combustion process in the industries [45,46,47]. The sorbents are typically used including zeolites, carbon–based materials, and aluminum formate (Al(HCOO)3) [48]. These sorbents not only effectively adsorb CO2 emissions but also demonstrate a remarkable capacity for selective CO2 capture with heightened efficiency [48, 49].”. Moreover, we are grateful for the previous recommendation of the three interesting references. For our study, we have chosen to cite two of them as they are directly relevant to our research.

Reviewer 4 Report

The authors properly addressed all my suggestions and concerns about the study. This versions is, from my perspective, suitable for publication.

Author Response

Thank you sincerely for your invaluable feedback on our manuscript. We deeply appreciate all of your helpful recommendations, which have significantly contributed to enhancing the quality of our work. Your insights have been instrumental in refining our manuscript to a more polished and improved version.